# Extended Applications of Small-Molecule Covalent Inhibitors toward Novel Therapeutic Targets

**DOI:** 10.3390/ph15121478

**Published:** 2022-11-27

**Authors:** Jesang Lee, Seung Bum Park

**Affiliations:** 1CRI Center for Chemical Proteomics, Department of Chemistry, Seoul National University, Seoul 08826, Republic of Korea; 2Department of Biophysics and Chemical Biology, Seoul National University, Seoul 08826, Republic of Korea

**Keywords:** covalent drugs, covalent inhibitors, targeted covalent inhibitors, drug discovery

## Abstract

Recently, small-molecule covalent inhibitors have been accepted as a practical tool for targeting previously “undruggable” proteins. The high target selectivity of modern covalent inhibitors is now alleviating toxicity concerns regarding the covalent modifications of proteins. However, despite the tremendous clinical success of current covalent inhibitors, there are still unmet medical needs that covalent inhibitors have not yet addressed. This review categorized representative covalent inhibitors based on their mechanism of covalent inhibition: conventional covalent inhibitors, targeted covalent inhibitors (TCIs), and expanded TCIs. By reviewing both Food and Drug Administration (FDA)-approved drugs and drug candidates from recent literature, we provide insight into the future direction of covalent inhibitor development.

## 1. Introduction

Covalent drugs have enhanced human lives by utilizing the covalent modification of target proteins. Aspirin, one of the most well-known anti-inflammatory drugs, and penicillin, one of the world’s first antibiotics, are the oldest examples of covalent drugs (Figure 1a) [1]. Historically, the discovery of covalent drugs has been serendipitous. Therefore, there was only a tiny window of opportunity for designing novel covalent inhibitors and exploring the covalent drug landscape. However, in recent decades, the rational design of covalent inhibitors has become more feasible and efficient with the advent of crystallography, computational chemistry, bioinformatics, and modern organic chemistry. One of the most pertinent examples is nirmatrelvir, one of two active components of Paxlovid (Figure 1a) [2]. This antiviral therapy is currently used to treat coronavirus disease 2019 (COVID-19), caused by the new virus, severe acute respiratory syndrome coronavirus 2 (SARS-CoV-2), which has intermittently evolved since its first outbreak in December 2019. When orally administered to COVID-19 patients, Paxlovid decreases mortality and hospitalization rates [3]. The main protease (M^pro^) of SARS-CoV-2, an enzyme essential for viral transcription and replication, is the target of nirmatrelvir. By rationally incorporating a nitrile moiety close to the catalytic residue of M^pro^, previous peptidomimetic leads can be utilized to produce a more potent and selective covalent inhibitor of this viral enzyme. However, despite their therapeutic potential, covalent drugs currently make up 4.4% of those approved by the U.S. Food and Drug Administration (FDA) within the past decade (Figure 1b) [4,5,6]. Of these, 90% serve as anticancer or antibiotic agents (Figure 1c).

Covalent drugs have several advantages over their non-covalent counterparts; for example, they elicit higher binding affinity through covalent bonds, allowing a longer duration of action by decoupling pharmacodynamics from pharmacokinetics. However, target selectivity remains the biggest challenge for covalent inhibitor development [1,7,8,9,10]. Undesired chemical modification of off-target proteins may induce pathogenesis, such as hepatotoxicity by liver protein inhibition and idiosyncratic drug-related toxicity by protein haptenization. Targeting catalytic residues of pathological enzymes has been the conventional mechanism of covalent inhibitors, from aspirin to nirmatrelvir (Figure 2a). However, these traditional covalent inhibitors may have poor subtype-selectivity if target residues are highly conserved across the protein family members. To this end, Singh and co-workers proposed a new strategy using targeted covalent inhibitors (TCIs) in 2011 (Figure 2b) [9]. This approach achieves subtype-selectivity by targeting poorly conserved and non-catalytic residues. Ibrutinib is a representative example of TCIs (Figure 1a). It is a first-in-class BTK inhibitor approved for treating mantle cell lymphoma and chronic lymphocytic leukemia (CLL). Along with the clinical success of ibrutinib, utilizing TCIs has since become the most popular approach for developing covalent drugs.

Recently, extended applications of TCIs to novel therapeutic targets have emerged in the clinic (Figure 2c). Most FDA-approved TCIs selectively inhibit kinases through poorly conserved cysteine residues near ATP-binding pockets. However, targetable cysteine residues are not prevalent in the human proteome [11]. Therefore, targeting residues other than cysteine at enzyme active sites have been the focus of drug development to overcome this restricted application. For instance, lysine has garnered attention as a feasible target residue for covalent modification because of its prevalence in the human kinome [12,13]. In addition, targeting allosteric sites has also been studied as an alternative strategy for direct inhibition of enzyme active sites [14,15]. Furthermore, non-enzymatic targets such as protein–protein interactions (PPIs) are gaining more interest as an underexplored area for covalent inhibitors [16,17,18,19,20].

Herein, we focus on a systematic analysis of recent covalent inhibitors based on their inhibition strategies: conventional covalent inhibitors, TCIs, and expanded TCIs (Figure 2). Target-based studies on the latest covalent inhibitors, especially kinase inhibitors targeting non-catalytic residues and protease inhibitors engaging catalytic residues, have already been well-addressed in other review articles [9,11,21,22,23,24]. In addition, comprehensive reviews on emerging covalent warheads have been reported by Gehringer and our group [25,26]. In this review, if available, representative examples of FDA-approved drugs and drug candidates are compiled for each inhibition strategy. We also discussed emerging covalent therapeutics, such as anti-inflammatory agents, and classical therapeutic indications, including anticancer and antibiotic drugs. Ultimately, we intended to provide insight into the potential therapeutic area where covalent drugs can be applied.

## 2. Recent Studies on Covalent Inhibitors

An updated overview of covalent inhibitors collected from a list of FDA-approved drugs and other articles is shown in Table 1, Table 2 and Table 3. We categorized covalent inhibitors based on their inhibition strategy: conventional covalent inhibitors, TCIs, and expanded TCIs. Chemical structures, target proteins, therapeutic indications, and warheads were also described in further detail. Over the past decade, enzyme inhibitors have been the most popular class of covalent inhibitors. For example, all three covalent drugs approved by the FDA last year—sotorasib, mobocertinib, and Paxlovid—were enzyme inhibitors. More recently, novel therapeutic targets other than enzymes, such as PPIs, receptor-ligand interactions, or transcription factors, have increasingly drawn attention. As a result, covalent inhibitors have become more structurally diverse, varying from ibrutinib with its BTK-targeting acrylamide warhead to voxelotor and its sickle cell hemoglobin-inhibiting salicylaldehyde group.

### 2.1. Conventional Covalent Inhibitors

Conventional covalent inhibitors, often called mechanism-based or suicide inhibitors, target the catalytic machinery of enzymes [27]. Aspirin and penicillin belong to this group and are the oldest examples of covalent inhibitors. In the case of proteases, serine, threonine, and cysteine are essential catalytic residues for their proteolytic function [24]. Targeting these catalytic residues may result in low selectivity as they are less prone to mutation because of their critical roles in enzyme function and are thus highly conserved across the protein family [7]. However, we can achieve selective cytotoxicity toward pathogenic cells depending on the cellular state or whether the target is restricted. For instance, penicillin selectively kills bacteria, not host cells, because bacterial proteases are absent in the human proteome. Another example is bortezomib (Table 1), the human proteasome inhibitor that selectively induces apoptosis in leukemia cells because of the higher dependency of cancer cells on the proteasome [28].

**Table 1 pharmaceuticals-15-01478-t001:** Representative examples of conventional covalent inhibitors.

Name/Structure	Target(s)	Therapeutic Indication	Warhead	Ref. (*Approval Date*)
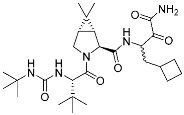 Boceprevir	Viral protease (*HCV NS3*)	Antiviral (*hepatitis*)	α-Ketoamide	*(13 May 2011)*
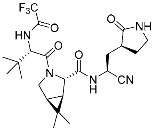 Nirmatrelvir (**1**)	Viral protease (*SARS-CoV-2 M^pro^*)	Antiviral (*COVID-19*)	Nitrile	[2] (*Emergency use authorization*, *22 December 2021)*
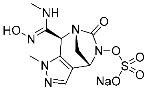 ETX0462 (**2**)	Bacterial protease (*PBP3, β-lactamase*)	Antibiotic (*MDR infection*)	Diazabicyclooctane	[29]
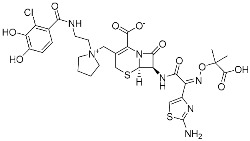 Cefiderocol	Bacterial protease (*PBP*)	Antibiotic (*MDR infection*)	β-Lactam	(*14 November 2019)*
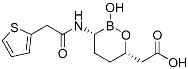 Vaborbactam	Bacterial protease (*β-lactamase*)	Antibiotic (*MDR infection*)	Cyclic boronic acid	(*29 August 2017)*
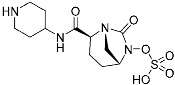 Relebactam	Bacteiral protease (*β-lactamase*)	Antibiotic (*MDR infection*)	Diazabicyclooctane	(*16 July 2019)*
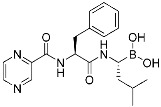 Bortezomib	Human proteasome	Anticancer (*leukemia*)	Boronic acid	*(13 May 2003)*
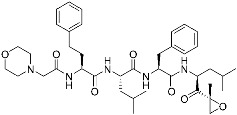 Carfilzomib	Human proteasome	Anticancer (*multiple myeloma*)	Epoxy ketone	*(20 July 2012)*
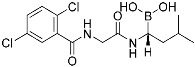 Ixazomib	Human proteasome	Anticancer (*multiple myeloma*)	Bornoic acid	*(20 November 2015)*
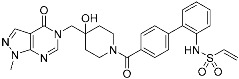 FT827 (3)	Deubiquitinase (*USP7*)	Anticancer	Vinyl sulfonamide	[30]
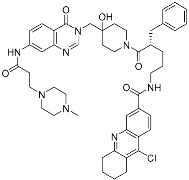 XL177A	Deubiquitinase (*USP7*)	Anticancer	Chlorotetrahydroacridine	[31]
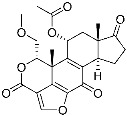 Wortmannin	Kinase (*PI3K*)	N/A	Furan	[1]
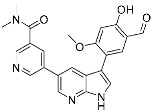 A5 (4)	Kinase (*BCR-ABL*)	Anticancer (*chronic myeloid leukemia*)	Aldehyde	[13]

Viral proteases have entered the spotlight following the successful development of Pfizer’s oral COVID-19 antiviral therapeutic, Paxlovid. However, only a few covalent inhibitors have been translated into clinics. Among them is boceprevir (Victrelis^TM^, Merck), an inhibitor of hepatitis C virus non-structural protein 3 (HCV NS3) (Table 1) [32]. The α-ketoamide group of Victrelis covalently binds to the active site Ser139 of HCV NS3. However, Merck discontinued Victrelis because there is a more potent anti-HCV drug, ledipasvir-sofosbuvir (Harvoni^TM^, Gilead). The latest successful covalent antiviral agent is Paxlovid, a combination of nirmatrelvir and ritonavir (Figure 3a). Among other chemical entities, Paxlovid is still the only FDA-approved drug that inhibits the main protease of SARS-CoV-2 [33,34,35,36,37]. Nirmatrelvir (**1**), the key component that stops viral replication, has an unconventional nitrile warhead that forms a covalent bond with Cys145 of the SARS-CoV-2 main protease (Figure 3a). Ritonavir is used to slow down the metabolic degradation of nirmatrelvir. The original lead compound of nirmatrelvir was reported in 2002 as a mimic of a peptide that binds to the SARS-CoV-1 main protease [2]. Therefore, the rational incorporation of a covalent warhead into the lead compound has enabled selective inhibition of newly emerging virus variants.

More examples of conventional covalent inhibitors are found in the field of antibiotics. Penicillin was discovered in 1929 to inhibit the penicillin-binding proteins (PBPs), which have transpeptidase activity and participate in bacterial cell wall synthesis. However, bacterial β-lactamases (BLAs), the enzymes responsible for breaking down β-lactam rings, have proven to induce primary resistance against penicillin therapy [38]. Consequently, there has been an urgent need for novel antibiotics to overcome multi-drug resistance (MDR). Tommasi and co-workers reported a novel covalent antibiotic, ETX0462 (**2**), which inhibits multiple PBPs and BLAs (Figure 3b) [29]. Using a rational design approach, they successfully targeted PBP1a and PBP3 using the diazabicyclooctane scaffold. Notably, for PBP3, the covalent modification of catalytic Ser294 through ring opening was suggested by the X-ray crystal structure. The inhibition of BLAs was also observed without covalent modification. High in vivo efficacy was achieved by improving bacterial porin permeability. Three covalent drugs to treat MDR infection were FDA-approved in the past five years: cefiderocol, vaborbactam, and relebactam (Table 1). Among them, cefiderocol inhibited PBPs with resistance to β-lactamase. Vaborbactam and relebactam were approved as β-lactamase inhibitors.

While pathogenic proteases, such as viral and bacterial proteases, have been investigated as targets of covalent inhibitors, many recent studies have characterized the covalent inhibition of mammalian or host proteases. Proteasome inhibitors are a well-established example of this class of drugs. To date, three covalent proteasome inhibitors have been FDA-approved for the treatment of multiple myeloma: bortezomib (Velcade^TM^, Millennium), carfilzomib (Kyprolis^TM^, Onyx), and ixazomib (Ninlaro^TM^, Takeda). Deubiquitinase (DUB) inhibitors have also been identified as a novel class of anticancer agents [24]. Komander, Kessler, and co-workers reported covalent inhibitors of ubiquitin-specific protease 7 (USP7), one of the DUBs implicated in the immune surveillance in the tumor microenvironment (Figure 3c) [30]. Using the ubiquitin-rhodamine assay, they discovered several USP7 inhibitors, including covalent inhibitor FT827 (**3**). According to the co-crystal structure, FT827 trapped an inactive apo-state of USP7, and its vinyl sulfonamide moiety formed a covalent bond with the catalytic Cys223 of USP7. Based on the pharmacophore of FT827, Buhrlage and co-workers developed XL177A by incorporating chlorotetrahydroacridine warheads (Table 1) [31] and demonstrated that XL177A selectively inhibits USP7 across the human proteome. The treatment of XL177A induces the degradation of MDM2, one of the substrates of USP7, followed by the increased level of p53, resulting in the inhibition of cancer cell growth.

As feasible targets of conventional covalent inhibitors, human kinases possess highly conserved catalytic lysine residues in their active sites. The representative example is Wortmannin, the covalent PI3K inhibitor targeting catalytic Lys833 (Table 1) [1]. However, lysine residues are challenging to react with covalent warheads because of their low nucleophilicity compared to cysteine [25,39,40]. In addition, targeting evolutionarily conserved lysine may result in poor target selectivity. To address these issues, there was a community effort to fine-tune the reactivity of covalent warheads and the selective recognition of non-covalent pharmacophores. Recently, Yao and co-workers reported novel covalent inhibitors of BCR-ABL kinase [12,13]. The fusion mutation of ABL kinase showed constitutive activity that resulted in leukemia. Previously, they utilized carbonyl boronic acid to improve the potency against drug-resistant ABL mutants. However, they observed poor cellular activity compared to that observed in the in vitro kinase assay. Alternatively, they incorporated different classes of warheads, SuFEx or salicylaldehyde moieties, within a non-covalent inhibitor, PPY-A (Figure 3d). The co-crystal structure of ABL kinase showed that salicylaldehyde-based inhibitor A5 (**4**) covalently bound to catalytic Lys271 at its active site (Figure 3d). They also demonstrated that A5 selectively targets the ABL and its mutant kinases confirmed by human kinome analysis. In the cellular assay, A5 successfully inhibited most cancer cells containing drug-resistant BCR-ABL mutants.

### 2.2. Targeted Covalent Inhibitors

Non-catalytic residues are another targetable site of covalent inhibitors since non-catalytic residues are poorly conserved across the protein family or target-restricted in contrast to chemically essential catalytic residues. Thus, covalent inhibitors targeting non-catalytic residues might have distinct selectivity compared to conventional covalent inhibitors. This alternative class of therapeutics called “targeted covalent inhibitors” (TCIs) was proposed by Singh and co-workers in 2011 [7]. As a result of persistent clinical translation of TCIs in the past decade, eight TCIs have been FDA-approved from 2012 to 2021, with several drug candidates currently under clinical evaluation (Table 2). Systematic sequence alignments have identified poorly conserved and non-catalytic residues as potential targets of covalent inhibitors. For example, Bruton’s tyrosine kinase (BTK) was determined to have non-catalytic Cys481, a residue rarely found in other human kinases [4]. This strategy resulted in the discovery of ibrutinib, the first FDA-approved BTK inhibitor for treating lymphoma, in 2013. Recently, two additional covalent BTK inhibitors with improved off-target selectivity received FDA approval: acalabrutinib and zanubrutinib (Table 2).

**Table 2 pharmaceuticals-15-01478-t002:** Representative examples of targeted covalent inhibitors.

Name/Structure	Target(s)	Therapeutic Indication	Warhead	Ref. (*Approval Date*)
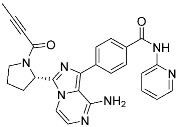 Acalabrutinib	Kinase (*BTK*)	Anticancer (*mantle cell lymphoma*)	2-Butyneamide	*(31 October 2017)*
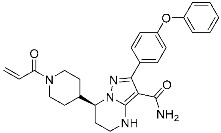 Zanubrutinib	Kinase (*BTK*)	Anticancer (*mantle cell lymphoma*)	Acrylamide	*(14 November 2019)*
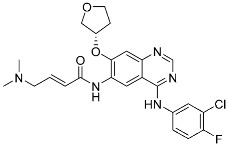 Afatinib	Kinase (*EGFR T790M* *and pan-HER*)	Anticancer (*NSCLC*)	Acrylamide	*(12 July 2013)*
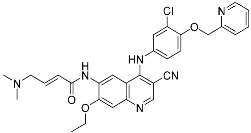 Neratinib	Kinase (*pan-HER*)	Anticancer (breast cancer)	Acrylamide	*(17 July 2017)*
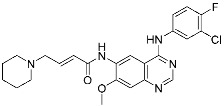 Dacomitinib	Kinase (*pan-HER*)	Anticancer (NSCLC)	Acrylamide	*(27 September 2018)*
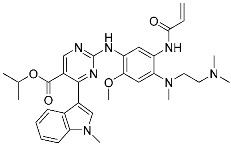 Mobocertinib (**5**)	Kinase (*EGFR ex20ins*)	Anticancer (*NSCLC*)	Acrylamide	[41] (*15 September 2021)*
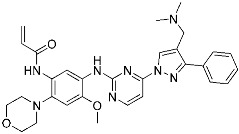 Lazertinib	Kinase (*EGFR*)	Anticancer (*NSCLC*)	Acrylamide	[42] (*Accerlated approval,* *21 May 2021, combination with amivantamab*)
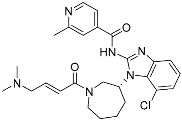 Nazartinib	Kinase (*EGFR*)	Anticancer (*NSCLC*)	Acrylamide	[42]
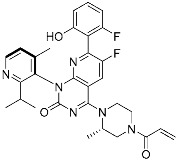 Sotorasib (**6**)	GTPase (*KRAS^G12C^*)	Anticancer (*NSCLC*)	Acrylamide	[14,43,44,45,46] (*28 May 2021)*
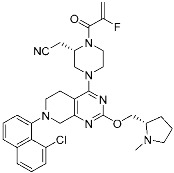 Adagrasib	GTPase (KRAS^G12C^)	Anticancer (NSCLC)	2-Fluoroacrylamide	[47] (*Under new drug application)*
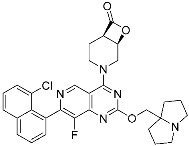 G12Si-5 (**7**)	GTPase (KRAS^G12S^)	Anticancer (NSCLC)	β-Lactam	[48]
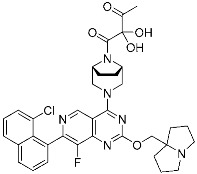 G12R inhibitor-4 (**8**)	GTPase (KRAS^G12R^)	Anticancer (NSCLC)	α,β-Diketoamide	[49]

Epidermal growth factor receptor (EGFR) is a member of the ErbB (HER) family composed of four receptor tyrosine kinases; EGFR (HER1, ErbB1), HER2 (ErbB2), HER3 (ErbB3), and HER4 (ErbB4). As ErbB protein family singaling regulates cellular growth and differentiation, its dysregulation is related to tumor development. EGFR has been the oldest target of TCIs as it has Cys797, which is poorly conserved across human kinases [4]. Afatinib is the first FDA-approved covalent EGFR inhibitor that targets lung cancer (Table 2). Afatinib inhibits the EGFR T790M gatekeeper mutant through a specific covalent bond, unlike previous non-covalent inhibitors [50]. Furthermore, afatinib covalently binds to Cys805 of HER2 and is called pan-HER inhibitor [4,21]. FDA-approved neratinib and dacomitinib also belong to this group (Table 2). However, afatinib showed dose-dependent cytotoxicity through inhibition of wild-type EGFR. Furthermore, EGFR exon 20 insertions (ex20ins), found in 9.1% of EGFR-mutant non-small cell lung cancer (NSCLC) patients, are insensitive to afatinib [42]. Heymach and co-workers have introduced an isopropyl ester group as a steric bump to occupy the selectivity hole in EGFRex20ins. As illustrated in Figure 4a, a conventional acrylamide warhead was utilized to engage non-catalytic Cys797 near the ATP-binding pocket. This approach resulted in the development of mobocertinib (**5**), which received FDA approval in 2021, as a therapeutic for NSCLCs with EGFRex20ins. Notably, more clinical successes of mutant-selective covalent EGFR inhibitors have been reported: lazertinib and nazartinib (Table 2) [42]. Meanwhile, HER3 inhibitor is an interesting example showing the applicability of TCI strategy. In contrast to other ErbB family members, HER3 is a pseudokinase that does not have the conserved Asp813 and Glu738 required for catalytic function. However, Gray, Crews, Jänne, and co-workers were able to develop the covalent inhibitor TX1-85-1 which targets HER3 through non-catalytic Cys721 in the ATP-binding pocket [51].

Recently, TCIs targeting non-catalytic residues on allosteric sites have been reported. In May of 2021, sotorasib (**6**) was FDA-approved for treating KRAS G12C mutation-derived NSCLCs through targeting allosteric Cys12, which is absent in wild-type KRAS (Figure 4b) [14,43,44,45,46]. KRAS is a GTPase that becomes inactivated upon converting bound GTP to GDP. Accordingly, capturing the switched-off state of GDP-bound KRAS is one promising strategy for selective target inhibition. Shokat and co-workers targeted allosteric Cys12 at the vicinity of switch-II pocket (S-IIP) to capture the GDP-bound form (Figure 4b). They found that the covalent bond between Cys12 and acrylamide warhead allowed sotorasib to bind tightly even at the shallow allosteric site of KRAS. Moreover, wild-type KRAS lacks a targetable cysteine residue, and sotorasib was shown to specifically inhibit G12C mutant KRAS. In addition to sotorasib, another chemical entity targeting KRAS G12C for NSCLC treatment is adagrasib, which is currently under the new drug application of the FDA (Table 2) [47].

Targetable residues of TCIs are not limited to cysteine. Nucleophilic residues other than cysteine can be selectively targeted via a proper introduction of covalent warheads in close proximity. Shokat and co-workers reported the covalent inhibitors (**7**, **8**) of KRAS G12S mutants and G12R mutants [48,49]. Based on the pharmacophore of adagrasib, they replaced 2-fluoroacrylamide warhead with the β-lactam moiety to target allosteric Ser12 near the S-IIP of the KRAS G12S mutant (Figure 4c). Similarly, by incorporating α,β-diketoamide warhead to the pyridopyrimidine moiety found in **7**, they successfully targeted the allosteric Arg12 of KRAS G12R mutants (Figure 4d).

### 2.3. Expanded Targeted Covalent Inhibitors

The covalent inhibition approach can be applied to non-enzymatic targets, including PPIs. PPIs are promising therapeutic targets, as they involve multiple cellular functions like transportation, structural supports, and immune responses [52]. Compared to previous antibodies and peptides targeting PPIs, small-molecule inhibitors have low molecular weights; thus, they can provide better pharmacokinetic profiles. Until now, PPIs were regarded as “undruggable” targets of small-molecule inhibitors since the interfaces between protein partners are generally large and shallow. In recent years, small-molecule TCIs have received growing attention as a promising strategy for PPI inhibition since TCIs targeting non-catalytic nucleophilic residues at PPI interfaces might overcome the unfavored binding thermodynamics of small molecules to their target proteins.

There is still a very limited clinical success with covalent PPI inhibitors; we can identify only one case over the past decade. As shown in Figure 5a, selinexor (**9**) was FDA-approved in 2019 for treating lymphomas and leukemia by covalently inhibiting the highly activated nuclear export receptor, exportin 1 (XPO1) [16]. XPO1 transports cargo proteins, such as growth-regulatory proteins, upon recognizing their nuclear export signal (NES). XPO1 has cysteine residue at its orthosteric site, and thus this can be a target of TCIs. To mediate XPO1 inhibition, selinexor covalently binds to Cys528 at the NES-binding groove of XPO1 through an acryloyl hydrazide warhead (Figure 5a).

However, PPI interfaces do not generally have targetable cysteine residue in an ideal position. Rather, the lysine residue is one of the most common residues on the protein surface as well as the PPI interface. Hence, lysine can be one possible alternative for the covalent modification with TCIs. Shown in the examples of covalent BCR-ABL kinase inhibitors, lysine-targeting warheads have been studied by many. Luo and co-workers discovered small-molecule inhibitors (**10**) for the PPI between LC3B and LIRs (LC3-interaction regions) (Figure 5b) [17]. They utilized enaminones as warheads to target Lys49 at the orthosteric site of LC3B. DC-LC3in-D5 compromised LC3B lipidation, thereby inhibiting the subsequent autophagy pathway in HeLa human cervical cancer cells. As of now, the clinical translation of DC-LC3in-D5 requires further in vitro and in vivo examination, as autophagy is closely related to various human diseases, including cancers, autoimmune diseases, and neurodegenerative diseases.

Hamachi and co-workers reported small-molecule inhibitors (**11**) for the HDM2/p53 PPI by modifying either tyrosine or N-terminal amine of target proteins (Figure 5c) [18]. In fact, the selective modulation of PPI between HDM2 and p53 renders a promising therapeutic target for several types of cancers. These inhibitors contain novel aryl sulfonyl fluorides, *N*-acyl-*N*-alkyl sulfonamides (NASAs), as a covalent warhead for the HDM2/p53 PPI. It was previously reported that NASAs efficiently reacted with the amino group of a non-catalytic lysine residue of Hsp90. Here, the NASA group was incorporated in nutlin-3, the first-in-class reversible HDM2 inhibitor that binds at the interface of HDM2/p53 PPI. Upon treatment of NASA-attached nutlin-3, the *N*-terminal α-amine and Tyr67 of HDM2, residues both seldom targeted by TCIs, were covalently modified (Figure 5c). This molecule was originally designed to target Lys51 of HDM2, but it unexpectedly modified other residues, including *N*-terminal α-amine and Tyr67. The covalent modification of HDM2 prolonged the activation of p53 and effective induction of the p53-mediated cell death compared to nutlin-3, a non-covalent inhibitor. Because Roche’s phase III trial with optimized nutlin-3, idasanutlin, failed in the treatment of acute myeloid leukemia (AML) in 2020, the clinical validation of NASA-attached nutlin-3 remains to be further investigated [53].

Allosteric covalent inhibition is another method for inhibiting PPIs. The extended TCIs discussed so far in this review all target orthosteric sites of protein partners. However, there may be no nucleophilic residues within the binding interface between protein partners. Instead, proteins may have an allosteric pocket containing nucleophilic residues, such as cysteine and lysine. One way to allosterically regulate proteins of PPIs is protein palmitoylation [20,54]. For example, TEAD/Yap1 interactions are enhanced by cysteine palmitoylation at the allosteric pocket of TEAD, which is located away from its orthosteric site [19]. TEAD/Yap1 interactions are associated with cancer in that they are involved in the Hippo signaling pathway, which regulates tissue homeostasis and organ size. Nonetheless, the interface of TEAD/Yap1 interaction is large and lacks a well-defined druggable binding pocket, making it difficult to achieve the desired PPI inhibition. Meroueh and co-workers were the first to discover allosteric covalent inhibitors (**12**) for the TEAD/Yap interaction (Figure 5d). They identified α-chloromethyl ketone-based inhibitors that covalently modify Cys367 within the palmitate binding pocket of TEADs (Figure 5d). However, some covalent binding analogs failed to inhibit the TEAD/Yap1 interactions, implying that the formation of the covalent bond itself is not sufficient for PPI inhibition. Another pair of protein partners whose interaction is allosterically regulated by protein palmitoylation is STING (stimulator of interferon genes) and either TBK1 or IRF3 (Table 3). Ablasser and co-workers reported covalent inhibitors targeting Cys91 at the palmitoylation sites of mouse STING with nitrofuran as a covalent warhead [55]. Cysteine palmitoylation of STING results in its translocation from the endoplasmic reticulum (ER) to the Golgi, which in turn induces interactions with TBK1 and IRF3. Given its crucial role in activating the intracellular DNA sensing pathway as innate immune responses, and in autoimmune diseases, the modulation of STING by allosteric covalent inhibition holds important clinical implications.

**Table 3 pharmaceuticals-15-01478-t003:** Representative examples of expanded targeted covalent inhibitors (or modulators).

Name/Structure	Target(s)	Therapeutic Indication	Warhead	Ref. (*Approval Date*)
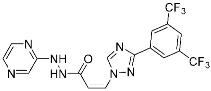 Selinexor (**9**)	Protein–protein interaction (*XPO1 and NES*)	Anticancer	Acrylamide	[16] (*3 July 2019*)
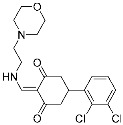 DC-LC3in-D5 (**10**)	Protein–protein interaction (*LC3B and LIR*)	N/A	Enaminones	[17]
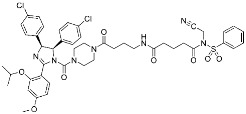 Covalent Nutlin-3 (**11**)	Protein–protein interaction (*HDM2and p53*)	Anticancer	*N*-Acyl-*N*-alkyl sulfonamides	[18]
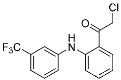 TED-347 (**12**)	Protein–protein interaction (*Yap1 and TEAD4*)	Anticancer	Chloromethyl ketone	[19]
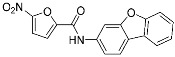 C-178	Membrane protein (*mouse STING*)	Anti-inflammatory drugs (*autoimmune disease*)	Nitrofuran	[55]
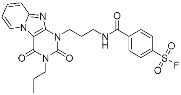 A_3_AR inhibitor **17b**	Membrane protein (*GPCR hA_3_AR*)	Glaucoma, Asthma	Fluorosulfonyl	[56]
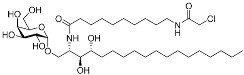 α-galactosylceramides	Membrane protein (*CD1d*)	Anti-inflammatory drugs	Chloroacetamide	[57]
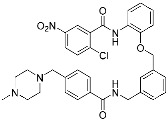 SB1453	Transcription factor (*PPARγ*)	Anti-diabetic (*type II diabetes*)	2-chloro-5-nitrobenzamide	[58]
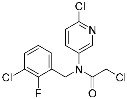 BPK-26	Transcription factor (*NR0B1*)	Anticancer	2-Chloroacetamide	[59]
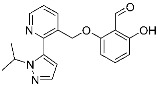 Voxelotor	Sickle cell hemoglobin	Sickle cell disease	Aldehyde	[60] *(25 November 2019)*

## 3. Summary and Perspectives

Here, we systematically reviewed the recent covalent inhibitors based on their mechanism of target inhibitions: conventional covalent inhibitors, TCIs, and expanded TCIs. The appropriate strategy should be selected depending on the structural landscape of the target proteins; for instance, the existence of targetable residues and druggable pockets. As each strategy has its own limitations, the above three strategies have been used in a cooperative manner to cover more therapeutic targets. As a result, covalent drug development has now been accepted as a practical methodology in current drug discovery. This trend has been supported by cumulative FDA approvals of covalent drugs and clinical candidates.

In the future, non-enzymatic proteins are expected to be the novel therapeutic targets of covalent drugs, including PPIs, transport proteins, membrane proteins, and transcription factors (Table 3). For instance, A_3_AR inhibitor **17b** binds to human GPCR through sulfonyl fluoride and BPK-26 inhibits transcription factor NR0B1 using chloroacetamide [56,59]. In our lab, we have reported SB1453 that blocks obesity-induced phosphorylation of PPARγ for treatment of type II diabetes [58]. Covalent modulators instead of inhibitors also have received attention as a new class of covalent drugs. One representative example is voxelotor, a covalent drug for treating sickle cell disease [60]. Voxelotor is a conformation stabilizer and aggregation inhibitor of mutant hemoglobin (HbS). It prevents the polymerization of deoxy-HbS by allosterically increasing HbS affinity to oxygen. Covalent α-galactosylceramides to modulate lipid antigen-presenting protein CD1d was also reported [57]. Meanwhile, the extension of targetable proteins has been spurred by the advancement of chemoproteomic techniques [50]. In contrast to the structure-based design, rapid target identification and selectivity profiling are enabled by proteome analysis. The discovery of ARS-853, which is further optimized as sotorasib, exemplifies the power of the electrophile-first screening method using mass spectrometry. Together, we expect that more clinical success will be achieved in the next decade regarding the discovery of covalent drugs toward currently underexplored therapeutic targets.

## Figures and Tables

**Figure 1 pharmaceuticals-15-01478-f001:**
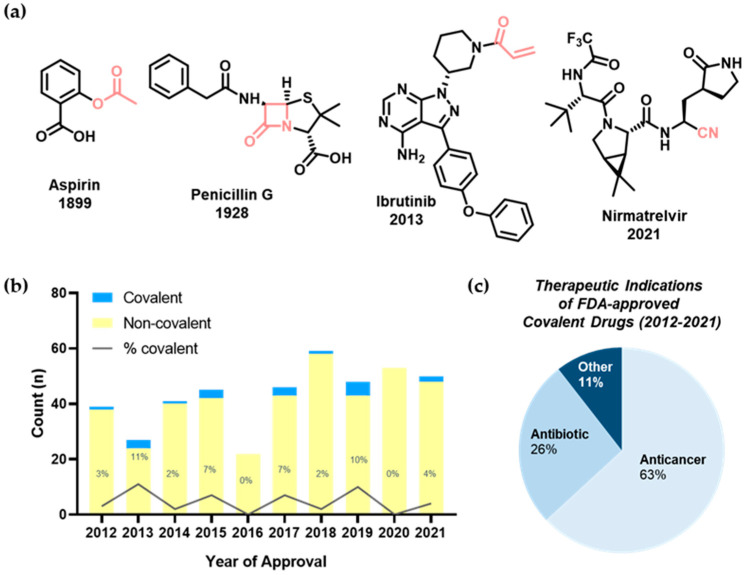
(**a**) Chemical structures of representative covalent inhibitors in early and recent years. (**b**) Overview of small-molecule covalent inhibitors in past decades (FDA approvals 2012–2021). The chart shows novel small-molecule covalent inhibitors among FDA approvals including new molecular entities (NMEs) and biologics license applications (BLAs). (**c**) Therapeutic indications of covalent inhibitors (total *n* = 19); anticancer (63%, *n* = 12), antibiotic (26%, *n* = 5), other (11%, *n* = 2).

**Figure 2 pharmaceuticals-15-01478-f002:**
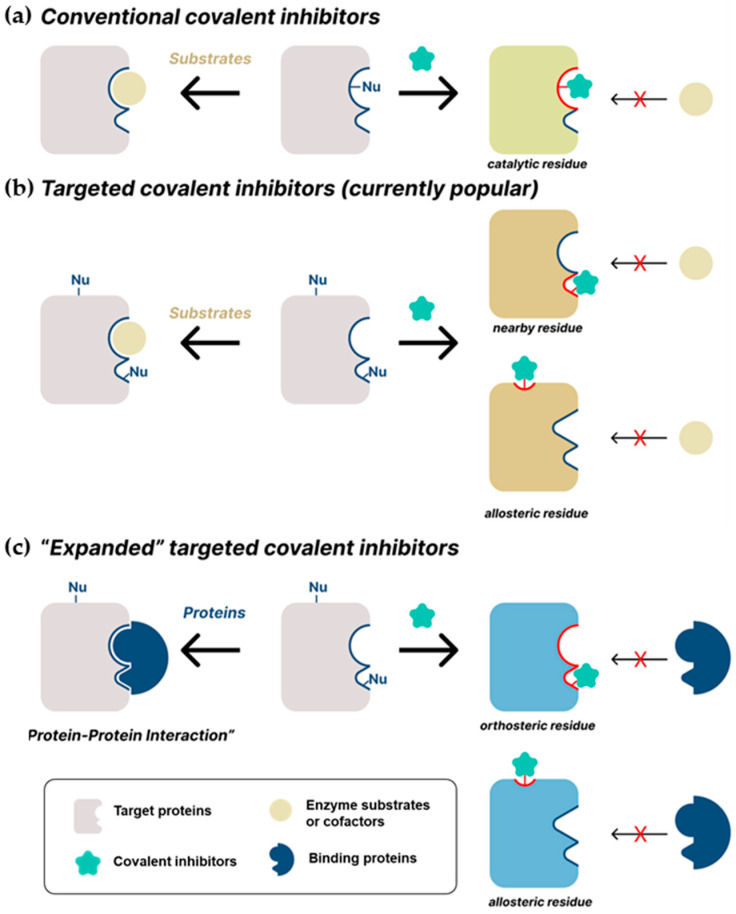
Schematic representation of the inhibition of target proteins (or interactions) with covalent inhibitors: (**a**) conventional covalent inhibitors; (**b**) TCIs; (**c**) expanded TCIs.

**Figure 3 pharmaceuticals-15-01478-f003:**
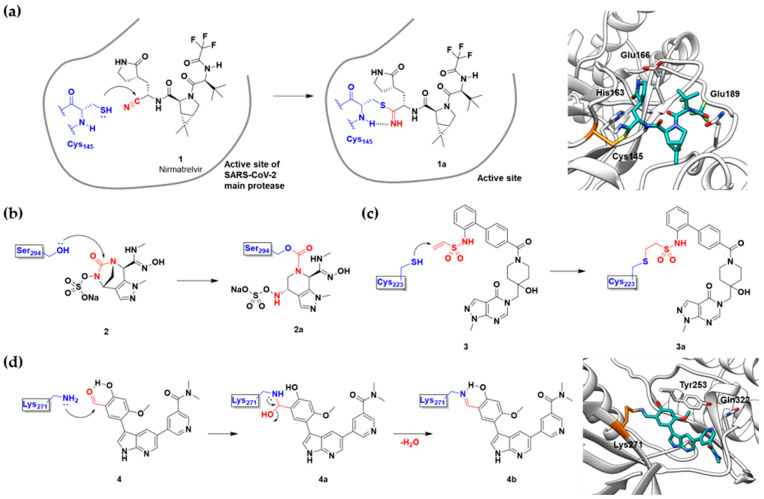
Representative examples of conventional covalent inhibitors: (**a**) schematic diagram and co-crystal structure of SARS-CoV-2 main protease with covalent inhibitor **1** (nirmatrelvir) (PDB: 7RFW). Nirmatrelvir is bound to the catalytic cysteine residue of the protease active site. Mechanism of the reaction between cysteine and nitrile warhead of nirmatrelvir generating thioimidate adduct **1a** is shown. Hydrogen bonding between cysteine α-amine and thioimidate stabilizes the covalent complex; (**b**) mechanism of the reaction between serine and diazabicyclooctane warhead of the covalent inhibitor **2** generating carbamate adduct **2a**; (**c**) mechanism of the reaction between cysteine and vinyl sulfonamide warhead of the covalent inhibitor **3** generating thioethane sulfonamide adduct **3a**; (**d**) schematic diagram and co-crystal structure of BCR-ABL kinase with covalent inhibitor **4** (PDB: 7W7Y). Mechanism of the reaction between lysine and salicylaldehyde warhead of **4** generating imine adduct **4a**. Hydrogen bonding between imine and phenol stabilizes the covalent complex.

**Figure 4 pharmaceuticals-15-01478-f004:**
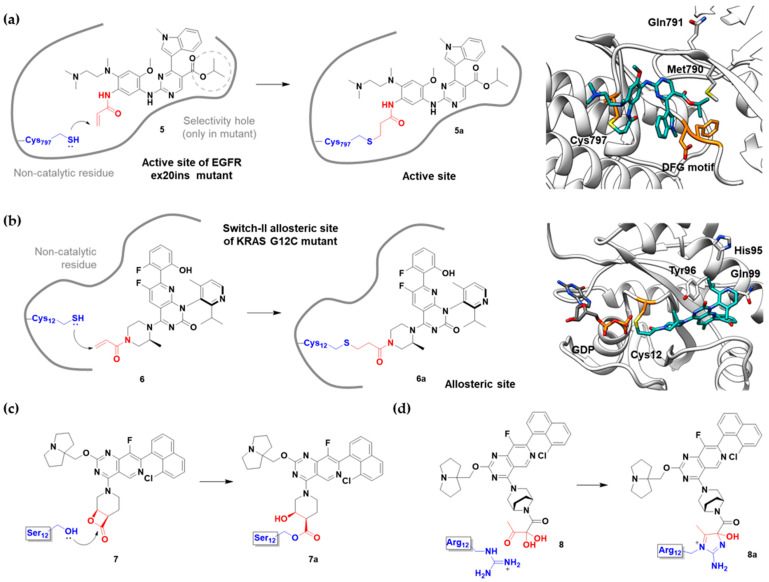
Representative examples of targeted covalent inhibitors: (**a**) schematic diagram and co-crystal structure of EGFR ex20ins mutant with covalent inhibitor **5** (mobocertinib) (PDB: 7A6K). Mobocertinib is bound to the non-catalytic cysteine residue of the kinase active site. Mechanism of the reaction between cysteine and acrylamide warhead of mobocertinib generating β-thiopropanamide adduct **5a** is shown. Only mutant kinase, which has a selectivity hole in contrast to wild-type, can accommodate the bulky isopropyl group of mobocertinib; (**b**) schematic diagram and co-crystal structure of KRAS G12C mutant with covalent inhibitor **6** (sotorasib) (PDB: 6OIM). Sotorasib is bound to the non-catalytic residue at the kinase allosteric site, switch-II pocket. Mechanism of the reaction between cysteine and acrylamide warhead of sotorasib generating β-thiopropanamide adduct **6a** is shown. The cysteine residue only present in the mutant confers the selectivity; (**c**) mechanism of the reaction between serine and lactone warhead of the covalent inhibitor **7** generating ester adduct **7a**; (**d**) mechanism of the reaction between arginine and α,β-diketoamide warhead of the covalent inhibitor **8** generating 4H-imidazolium adduct **8a**.

**Figure 5 pharmaceuticals-15-01478-f005:**
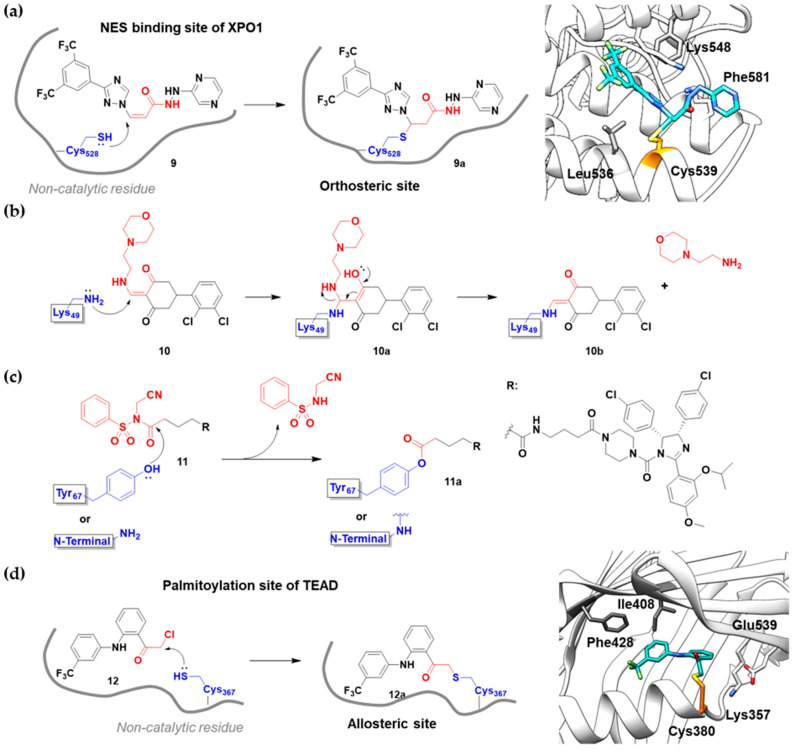
Representative examples of “expanded” targeted covalent inhibitors: (**a**) schematic diagram and co-crystal structure of exportin 1 (XPO1) with covalent inhibitor **9** (selinexor) (PDB: 7L5E). Selinexor is bound to the non-catalytic cysteine residue at the PPI interface. Mechanism of the reaction between cysteine and acryloyl hydrazide warhead of selinexor generating β-thiopropane hydrazide adduct **9a** is shown; (**b**) mechanism of the reaction between lysine and enaminone warhead of the covalent inhibitor **10** generating enaminone adduct **10a**; (**c**) mechanism of the reaction between tyrosine or *N*-terminal α-amine and NASA warhead of the covalent inhibitor **11** generating ester or amide adduct **11a**; (**d**) schematic diagram and co-crystal structure of TEAD with covalent inhibitor **12** (PDB: 6E5G). **12** is bound to the non-catalytic residue at the allosteric site of PPI. Mechanism of the reaction between cysteine and α-chloromethyl ketone generating α-thiomethyl ketone adduct **12a** is shown.

## Data Availability

Data availability not applicable.

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
