# Peer review of "Extended Applications of Small-Molecule Covalent Inhibitors toward Novel Therapeutic Targets"

_pharmaceuticals, 2022, doi:10.3390/ph15121478_

Round 1
Reviewer 1 Report
Dear Authors,
The systematic review you presented is well-structured, the sections were drawn up with all the needed information and a good number of details.
The aim is clearly presented and successfully reached.
This reviewer could not detect any style or grammar issues, the Figures are clear and high quality. The table reports the right information.
This reading is smooth and the data reported are of great interest and well-selected and listed.
Author Response
We deeply appreciate the strong support and encouragement of referee #1.
Based on referee #1 and #2's comments, we revised the manuscript accordingly.
After these changes, this focused paper is much more informative and legible.
Thanks again!!
Reviewer 2 Report
Dear authors of the review “Extended applications of small-molecule covalent inhibitors toward novel therapeutic targets“, I have only several comments:
1. Line 29: I would recommend changing “an active component of Paxlovid“ to “one of two active components of Paxlovid“.
2. Line 38: can you instead provide a decade´s average?
3. Table 1: what order organizes substances within each category? I would recommend alphabetical order of targets. Subsequently, the same target order could be used in the following text. Moreover, I might recommend splitting table 1 into 3 separate tables for each group.
4. Lines 133–134: I recommend placing “Three covalent antibiotic … (Table 1).“at the end of the paragraph. BUT: a) relebactam and vaborbactam are not antibiotics per se. b) lines 145–147 do not seem true because cefiderocol won this race in 2019. Therefore, the paragraph should be more about this compound and its mechanism.
5. Line 139: “Tommasi and co-workers reported“ … why using a name that is not the first author (this name is listed in “et al.” part)? The same goes for lines 155, 161, 217, 228, etc. Check the whole manuscript.
6. Line 142: check the spelling of “diazabicylcooctane“.
7. Line 153: can you list both substances in table 1?
8. Figure 3: is it necessary to place Figure 2a into Figure 3? The same goes for Figures 4 and 5.
9. Lines 167, 172, 173, 174: colons and semicolons should be followed by a lowercase letter. Similarly, check figures 4 and 5.
10. Line 170: what attacks what? Warheads attack their targets, or is it the other way around? Check it, please. If you decide to change it, go through figure legends 3, 4, and 5.
11. Line 186: “kinases”… singular or plural? The same goes for line 192 (“inhibitors”).
12. Line 209: can you add some information on Acalabrutinib and Zanubrutinib at the end of the paragraph?
13. Line 222: can you add some information on Lazertinib and Nazartinib?
14. Line 255: “incorporated in sotorasib”… are you sure? The molecule (8) looks somewhat different than the compound (6).
15. Chapter 2.2: can you add a paragraph summarizing HER inhibitors?
16. Line 314: can you add a reference to “30 min”?
17. Figures: can you provide a higher resolution format?
18. Chapter 2.3: can you add some information on 17b, α-galactosylceramides, SB1453, and BPK-26?
19. Line 344: use the abbreviation “TCIs” solely.
20. Lines 354 and 355: is it necessary to refer to “Table 1” again?
I hope my comments could help you improve the review.
Author Response
Please see the attached point-by-point responses!!
We deeply appreciate referee #2 for his/her constructive comments.
Based on referee #2's comments, we revised the manuscript.
After these changes, our manuscript is much more informative and legible.
Thanks again!!
